# Breaking the Spurious Causality of Conditional Generation via Fairness Intervention with Corrective Sampling

**Junhyun Nam**                                                    *junh.nam@samsung.com*
*Samsung Advanced Institute of Technology (SAIT)\**

**Sangwoo Mo**                                                         *swmo@kaist.ac.kr*
*Korea Advanced Institute of Science & Technology (KAIST)*

**Jaeho Lee**                                                    *jaeho.lee@postech.ac.kr*
*Pohang University of Science and Technology (POSTECH)*

**Jinwoo Shin**                                                      *jinwoos@kaist.ac.kr*
*Korea Advanced Institute of Science & Technology (KAIST)*

**Reviewed on OpenReview:** *https://openreview.net/forum?id=VV4zJwLwI7*

## Abstract

To capture the relationship between samples and labels, conditional generative models often inherit spurious correlations from the training dataset. This can result in label-conditional distributions that are imbalanced with respect to another latent attribute. To mitigate this issue, which we call spurious causality of conditional generation, we propose a general two-step strategy. (a) Fairness Intervention (FI): emphasize the minority samples that are hard to generate due to the spurious correlation in the training dataset. (b) Corrective Sampling (CS): explicitly filter the generated samples and ensure that they follow the desired latent attribute distribution. We have designed the fairness intervention to work for various degrees of supervision on the spurious attribute, including unsupervised, weakly-supervised, and semi-supervised scenarios. Our experimental results demonstrate that FICS can effectively resolve spurious causality of conditional generation across various datasets.

## 1 Introduction

Deep generative models are capable of producing synthesized visual content that closely resembles the training data, but sometimes in undesirable ways. Similar to classifiers, generative models can be affected by dataset bias (Torralba & Efros, 2011; Buolamwini & Gebru, 2018) inherited from the training set, resulting in significantly fewer samples of minority groups compared to majority group samples (Zhao et al., 2018; Salminen et al., 2020; Jain et al., 2021). This group imbalance is usually amplified in the generated dataset compared to the original training dataset, as noted by Tan et al. (2020). The exacerbated imbalance in generated samples can lead to various fairness issues in downstream applications, such as underrepresentation of minority groups in content creation (Choi et al., 2020; Tan et al., 2020; Jalal et al., 2021) or harm to the classifier trained with generated samples (Xu et al., 2018; Sattigeri et al., 2018; Xu et al., 2019).

Conditional generative models face a different type of fairness concern due to the association of the input condition and spurious attributes in data generation. Specifically, the input-conditional distribution of generated data may be imbalanced with respect to some latent attribute, replicating the spurious correlation present in the training dataset. For instance, a BigGAN (Brock et al., 2019) model trained on the CelebA (Liu et al., 2015) dataset with the "blondness" attribute as a class label may generate severely imbalanced samples in terms of "gender," with the vast majority of samples generated under the blond class being female (see

---

*\*Work done at KAIST

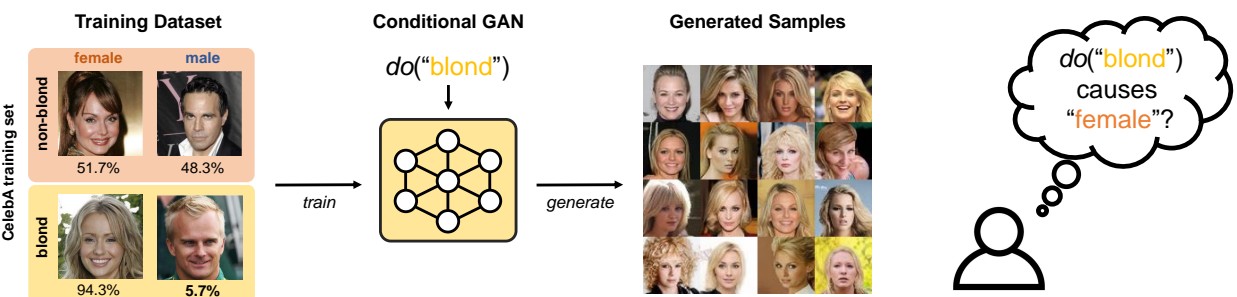

Figure 1: **Motivation.** Conditional generative models trained on datasets with spuriously correlated attributes may generate severely imbalanced samples when conditioned on the correlated attribute. This behavior is particularly problematic in conditional generation, as there is a causal relationship between the act of conditioning (intervention) and the distribution of generated samples, which can amplify biased beliefs held by users. We tackle this problem, which we refer to as spurious causality of conditional generation.

Figure 1). It has also recently been reported that popular text-to-image generation models exhibit gender and cultural biases (Barr, 2022). When prompted with specific occupations like "childcare worker," the generated images are highly imbalanced in terms of gender and race.

The reproduction of spurious correlations by conditional generative models raises a deeper fairness concern compared to imbalances in unconditional generative models. Input conditioning (i.e., providing the class label "blond") can be seen as an intervention from the perspective of counterfactual causality (Pearl, 2010). Once a spurious correlation is incorporated into conditional generative models, changing the input (intervention) leads to a change in the distribution of spuriously correlated latent attributes of the conditionally generated samples, forming a cause-effect relationship. This false sense of causation in conditional generation, which we refer to as *spurious causality of conditional generation*, is a significant ethical threat, as this misconception is likely to propagate when using generative models (Mariani et al., 2018).[1]

To the best of our knowledge, no previous work has aimed to address the spurious causality of conditional generation. Previous studies on fair generative models have focused on addressing majority bias by either (a) reducing the discrepancy from data distribution (Grover et al., 2019; Mo et al., 2019b; Lee et al., 2021; Yu et al., 2020; Humayun et al., 2022) or (b) following the desired fair distribution specified by explicit labels of sensitive attributes (Tan et al., 2020) or a reference set of desiderata (Choi et al., 2020). A naive solution to address the issue could be to extend the previous work on majority bias by applying it to each condition. Instead, we propose to leverage the conditional structure to tackle the problem.

**Contribution.** We propose a general framework, coined *fairness intervention with corrective sampling* (FICS), to mitigate the severe yet underexplored problem of spurious causality in conditional generative models. FICS consists of two components. (1) Fairness intervention (FI): emphasize samples suffering from the spurious correlation in the dataset at the training phase. (2) Corrective sampling (CS): explicitly filter samples after the training phase so that the trained generator follows the desired distribution.

FICS provides a unified approach to resolve spurious causality of conditional generation for a wide range of scenarios, with various degrees of supervision available on the latent attributes and various target ratio of latent attribute distributions. We focus on the task of class-conditional image generation using conditional generative adversarial networks (Mirza & Osindero, 2014, cGANs) and apply FICS on unsupervised, weakly supervised, and semi-supervised setups. The goal is to generate the conditional distributions whose latent attribute distribution is either identical to that of the training dataset or some desired fair distribution.

Our experiments show that FICS is an effective remedy for spurious causality of conditional generation. In particular, we train BigGAN (Brock et al., 2019) on CelebA (Liu et al., 2015) and Waterbirds (Sagawa et al., 2020), and ResNetGAN (Gulrajani et al., 2017) for Colored MNIST (Arjovsky et al., 2019). FICS consistently improves upon the conditional extensions of previous fair generation methods in both unsupervised and supervised scenarios, in terms of the sample quality and the minority attribute occurrence ratio.

---

[1]We use the term "spurious causality" as an analogy to the term "spurious correlation" used in classification contexts.

## 2 Related Work

**Generative models.** Generative models have synthesized high-fidelity samples in various domains, e.g., image (Karras et al., 2021), video (Yu et al., 2022), and language (Brown et al., 2020). Numerous methods have been proposed, including generative adversarial networks (Goodfellow et al., 2014, GANs) and diffusion models (Dhariwal & Nichol, 2021). Here, GANs are known to merit both sample quality and speed (Xiao et al., 2022). In addition, conditional GANs (cGANs) utilize additional supervision such as class labels (Mirza & Osindero, 2014), text descriptions (Ramesh et al., 2022; Ding et al., 2022), or reference images (Zhu et al., 2017; Mo et al., 2019a) as input conditions to extend the applicability further. Specifically, cGANs control the synthesized outputs and improve sample quality by restricting the output space for each condition. Despite their utility, we find that cGANs often suffer from a severe yet underexplored fairness issue. We investigate this issue, coined spurious causality of conditional generation, and propose a remedy for it.

**Fairness in generative models.** The fairness issue in generative models has received considerable attention due to its practical impact. Most prior work on fair generation focuses on addressing the majority bias, which is the tendency of generative models to learn the imbalances inherited from the training dataset and is often amplified during training (Tan et al., 2020). One line of work considers an unsupervised setup, where the group identities of samples are unknown. Approaches in this direction handle bias by using discrepancy metrics corrected with density ratios as the optimization objective (Grover et al., 2019; Mo et al., 2019b; Lee et al., 2021) or by imposing regularizations to enforce a uniform learning of the data manifold (Yu et al., 2020; Humayun et al., 2022). Another line of work learns the fair distribution in a supervised manner (Xu et al., 2018; Sattigeri et al., 2018; Xu et al., 2019; Tan et al., 2020; Choi et al., 2020). However, previous works only focus on the majority bias, and the spurious correlation over multiple attributes has not been investigated.

**Spurious correlation of classifiers.** Spurious correlations in training datasets can cause classifiers to perform poorly on out-of-distribution test sets where these correlations do not hold (Geirhos et al., 2020). To mitigate spurious correlation, several works have been proposed, such as invariant learning (Arjovsky et al., 2019) and distributionally robust optimization (Sagawa et al., 2020), but these methods require explicit supervision on the spuriously correlated attribute, which is often expensive and time-consuming to obtain. Weaker forms of supervision, such as fair validation sets (Bahng et al., 2020; Nam et al., 2020; Creager et al., 2021; Liu et al., 2021), a small set of attribute-annotated samples (Nam et al., 2022; Sohoni et al., 2021), or pre-trained vision-language models (Kim et al., 2023), have been explored to identify and mitigate poorly performing samples. Specifically, this line of work focuses on identifying and mitigating poorly performing samples. Our method shares the same philosophy as these debiasing methods for spurious correlation but instead focuses on tackling the spurious causality issue in conditional generative models.

**Casuality in generative models.** While our paper focuses on achieving fairness between multiple attributes in the vision domain, research has also been conducted in the tabular domain with a similar focus. In the tabular domain, each data point typically has all attribute labels, including the sensitive attribute of interest, making it easier to directly observe all attributes (Xu et al., 2018). Moreover, some research assumes prior knowledge of the causal structure among the attributes and leverages the parent attribute to generate its child attribute (Xu et al., 2019; van Breugel et al., 2021). However, our study addresses more flexible scenarios with varying levels of supervision on the sensitive attribute.

**Relation to counterfactual fairness.** Our framework shares some similarities with counterfactual fairness (Kusner et al., 2017) in terms of its ability to conduct counterfactual analysis on specific attributes. However, our framework performs the do-operation on the class attribute and focuses on the distribution of the sensitive attribute in the generated images. In contrast, counterfactual fairness aims to achieve fair prediction by performing the do-operation on the sensitive attribute. Therefore, the methods used to achieve counterfactual fairness are not directly applicable to addressing the problem of interest in our framework.

**Spurious correlations beyond class labels.** While we focus on the class attributes in this paper, spurious correlation also appears in different domains, e.g., keyword bias (Moon et al., 2021) or scene bias (Mo et al., 2021). It can be problematic for generative models using such conditions, e.g., text-to-image models can be biased to some keywords. Here, we remark that our principle of leveraging the spurious correlation of classifiers to mitigate the spurious causality of conditional generative models can be applied in general.

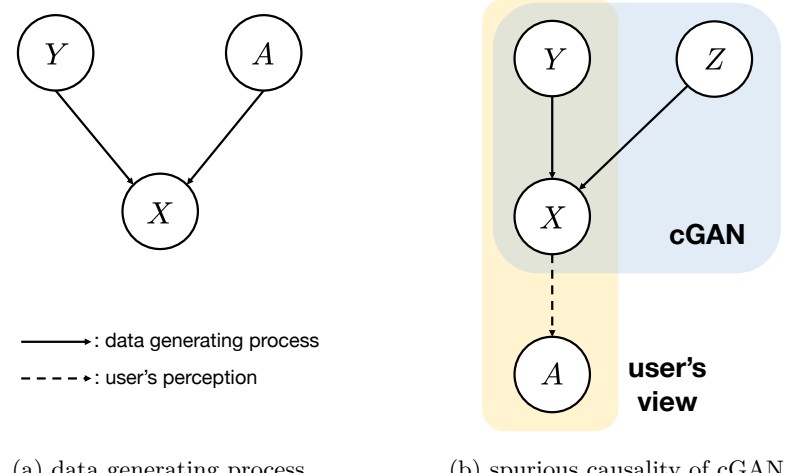

(a) data generating process      (b) spurious causality of cGAN

Figure 2: **Causal graphs.** Although there is no causal relationship between the class attribute $Y$ and the sensitive attribute $A$, conditioning the class to generate data can impact the sensitive attribute of the generated data. As a result, users of cGANs who observe the generated data and its sensitive attribute may falsely perceive that manipulating the class attribute alters the sensitive attribute of the generated data, as long as a spurious correlation exists between the class attribute and the sensitive attribute.

## 3   Spurious Causality of Conditional GAN

Conditional generative models aim to approximate the ground-truth distribution $p_{\mathsf{data}}(x|y)$ of data $x \in \mathcal{X}$ conditioned on a label $y \in \mathcal{Y}$ using a model distribution $p_{\mathsf{gen}}(x|y)$. For brevity, we assume that the models are conditional generative adversarial networks (Mirza & Osindero, 2014, cGAN). cGAN trains a generator $G : (z, y) \mapsto x$, which generates data $x$ conditioned on a latent variable $z \sim p(z)$ drawn from a fixed prior distribution $p(z)$, and a label $y \sim p_{\mathsf{data}}(y)$. This procedure defines the generator distribution $p_{\mathsf{gen}}(x|y)$. Simultaneously, cGAN trains a discriminator $D : (x, y) \mapsto r \in (0, 1)$, which predicts whether the data-label pair $(x, y)$ comes from the data distribution ($r = 1$) or the generator distribution ($r = 0$). Formally, the training objective of cGAN is:

$$\mathcal{L}_{\mathsf{cGAN}}(G, D) := \mathbb{E}_{y \sim p_{\mathsf{data}}(y)}[\mathbb{E}_{x \sim p_{\mathsf{data}}(x|y)} \log D(x, y) + \mathbb{E}_{z \sim p(z)} \log(1 - D(G(z, y), y))].$$

We consider training cGANs on a training dataset with spurious correlations. Specifically, we assume that there is a sensitive attribute $a(x) \in \mathcal{A}$ associated with the data $x$, which is correlated but not causally related to the label $y$. We observe that the model distribution $p_{\mathsf{gen}}(x|y)$, when naively trained, learns to replicate the spurious correlation present in the training dataset. More specifically, we define the label-conditional probability of sensitive attributes for the model distribution $p_{\mathsf{gen}}(x|y)$ as follows:

$$p_{\mathsf{gen}}(a|y) = \int_{\mathcal{X}} \mathbf{1}[a = a(x)]\, p_{\mathsf{gen}}(x|y)\, \mathrm{d}x$$

and define $p_{\mathsf{data}}(a|y)$ similarly. We observe that the model distribution $p_{\mathsf{gen}}(x|y)$ is trained in a way that $p_{\mathsf{gen}}(a|y)$ is significantly more imbalanced than $p_{\mathsf{data}}(a|y)$.

Such exaggerated reproduction of spurious correlation by conditional generative models is a big problem, as the model itself now provides a causal link between the act of "conditioning on the label" and the "sensitive attribute of the generated sample." Indeed, the model can be viewed as a black-box on which a counterfactual experiment can be performed by intervening on the label with everything else fixed (e.g., latent variable $z$). Such analysis will lead to a conclusion that there exists a causal structure (Pearl, 2010) between the label and the spurious attribute of the generated samples.

This problem, which we call *spurious causality of conditional generation*, is potentially more problematic than the spurious correlation learned by classifiers. Namely, conditional generative models can propagate

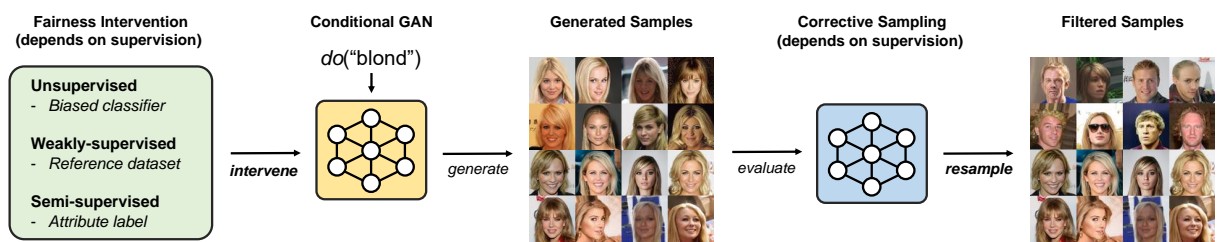

Figure 3: **Method overview.** The proposed FICS is a general two-stage framework that aims to address the spurious causality of conditional generation. It involves fairness intervention during the training phase to encourage the generator to synthesize minority samples (FI), followed by corrective sampling during the generation phase (CS). The specific steps depend on the type of fairness supervision.

harmful stereotypes, such as the belief that the attribute "female" is caused by being "blond," to users who directly observe the effect of providing conditions to the generated samples.

Our goal is to address the spurious causality of conditional generative models by learning a balanced conditional distribution $p_{\mathsf{gen}}(x|y)$ that yields a desired sensitive attribute distribution $p_{\mathsf{gen}}(a|y)$. The desired distribution may vary depending on the user intention:

(a) *Approximating the data distribution $p_{\mathsf{data}}(a|y)$.* As previously discussed, standard GAN training often yields a more imbalanced model distribution $p_{\mathsf{gen}}(a|y)$ than the data distribution $p_{\mathsf{data}}(a|y)$. Our aim is to learn a model distribution that closely approximates the data distribution, i.e., $p_{\mathsf{gen}}(a|y) \approx p_{\mathsf{data}}(a|y)$.

(b) *Approximating a fair distribution $p_{\mathsf{fair}}(a|y)$.* Our aim is to approximate a sensitive attribute distribution that aligns with specific fairness criteria. For instance, we may enforce the attribute to follow a fair distribution, i.e., $p_{\mathsf{gen}}(a|y) \approx p_{\mathsf{fair}}(a)$ for some desired $p_{\mathsf{fair}}(a)$.

In addition to achieving desired $p_{\mathsf{gen}}(a|y)$, we also consider achieving a *fair* generative quality for each label as an auxiliary goal. For this purpose, we also report the (fair) Intra-FID for the compared methods.

## 4 Breaking the Spurious Causality of Conditional GAN

### 4.1 Overall framework

We describe the proposed FICS (Figure 3), a general strategy for mitigating the spurious causality of conditional generation. In essence, FICS comprises two mechanisms to conditionally synthesize samples with desired attribute distributions, with each component being designed based on the level of supervision available on the latent attributes.

- *Fairness Intervention* (FI): Encourage the generator to synthesize minority samples during the training phase. The specific intervention rules depend on the degree of supervision available on the latent attributes.
- *Corrective Sampling* (CS): Apply rejection sampling on the synthesized samples during the generation phase. The criteria for rejection depend on the desired goal, follow the data distribution or fair distribution.

We consider two different scenarios and give concrete versions of FICS for each case. First, an unsupervised setup aims to recover the data distribution by fixing the bias of the current model. Second, a supervised setup aims to follow the desired fair distribution described by additional supervision, such as a reference set of desired distribution or labels of sensitive attributes. The following subsections illustrate how to design the fairness intervention and corrective sampling for unsupervised and supervised scenarios.

### 4.2 Unsupervised: Learning the data distribution

In the unsupervised scenario, we aim to reduce the bias in the generator by encouraging it to generate more samples that are likely to be mispredicted by a biased classifier. This approach is motivated by the

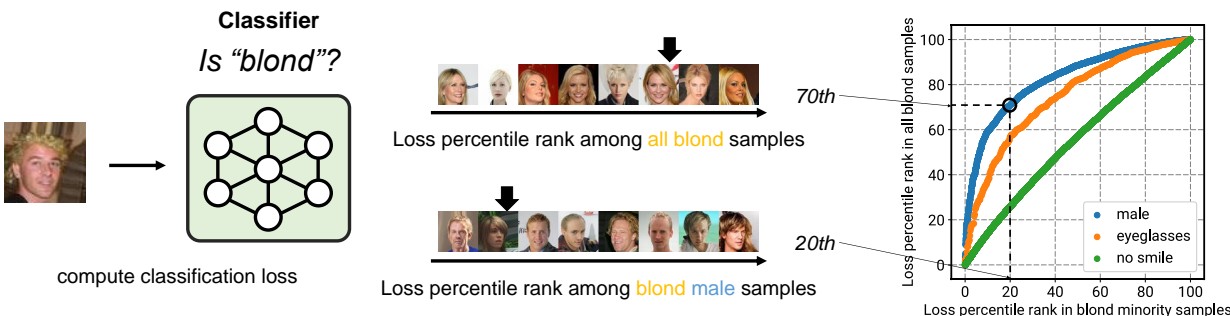

Figure 4: **Observation.** The relationship between the biased classifier and the generator is depicted by the percentile ranks of the classification loss of generated samples, with the x and y coordinates of a point representing the percentile rank of its classification loss among blond minority samples and all blond samples, respectively. For instance, a sample with a loss at the 20th percentile among blond male samples has a loss at approximately the 70th percentile among all blond samples, indicating that blond male samples typically have higher classification loss among all blond samples. The attributes of "gender" and "wearing eyeglasses" exhibit spuriously correlation with "blondness," while the attribute of "smiling" does not.

observation that there is a strong correlation between the misprediction of the biased classifier and the generation frequency of the biased cGAN. We first introduce this observation and then describe how to adjust the training and inference of the cGAN to mitigate this bias.

### 4.2.1 Observation

Our observation (Figure 4) suggests the usefulness of having a separately trained classifier to mitigate the spurious correlation of conditional generative models. Specifically, we train a classifier and a conditional generative model on the CelebA (Liu et al., 2015) dataset and focused on the relationship between the "blondness" attribute and the "gender," "wearing eyeglasses," and "smiling" attributes. When conditioned on blond, the dataset is severely imbalanced with respect to the gender and wearing eyeglasses attributes. Unlike the other two attributes, there is no clear correlation between blondness and smiling.

From the trained model, we observe that the generated samples classified as the minority group are more likely to have a higher classification loss. For example, given a blond-conditionally generated sample, a sample with a higher classification loss is more likely to be male. Specifically, we observe the percentile rank of the given sample among all blond samples and specific blond male samples. For instance, a sample having the 20th percentile among blond male samples has a loss with around the 70th percentile among all blond samples. We observed a similar pattern for the "wearing eyeglasses" attribute, which is spuriously correlated with the "blondness" attribute. In contrast, the "smile" attribute, which has no clear correlation with "blondness" attribute, does not show a distinct loss distribution for all blond samples and blond and no smile samples. This observation suggests that the classification loss of generated samples may contain useful information to identify minority group samples, even in cases where the generator and classifier are trained separately and receive no supervision about the correlated attributes (e.g., "gender," "wearing eyeglasses").

### 4.2.2 Training and sampling

**Fairness intervention.** Motivated by our observation, we intervene in the generator $G$ to encourage the synthesis of samples that are difficult for the biased classifier $f_b : x \mapsto y$ to classify. These samples are likely to have minor attributes, i.e., low $p(a|y)$. To achieve this, we apply a regularizer that encourages the generator to minimize the (sum of) cross-entropy loss over the wrong targets $y' \neq y$, rather than maximizing the correct target to ensure the convexity. This approach encourages the generator to generate more minority samples

that are hard to classify. Formally, our objective is:

$$\min_G \max_D \ \mathcal{L}_{\mathsf{cGAN}} + \lambda \cdot \sum_{y' \neq y} \mathsf{CE}(f_b(G(z,y)), y'), \tag{1}$$

where $\lambda$ is a weight for the regularizer and $\mathsf{CE}$ denotes the cross-entropy loss. We simply set $\lambda = 0.01$ since it worked well for all our experiments. We pretrain the classifier $f_b$ and fix it for the training of cGAN.

**Corrective sampling.** In the unsupervised scenario, the regularizer used in the first step can distort the training distribution. Therefore, we use discriminator rejection sampling to correct the distribution of the generated samples. Although the regularizer in Eq. (1) increases the weight of the minority samples, it does not guarantee that the generative models follow the data distribution. Thus, we apply correction sampling to recover the original data distribution.

Rejection sampling is a commonly used technique to sample from a probability distribution $p$ when the ratio $p/q$ is known and sampling from $q$ is feasible. This technique involves sampling $x$ from $q$ and accepting the samples with a probability proportional to $p/q$. This procedure is equivalent to sampling from $p$. In our scenario, we sample $x$ from $p_{\mathsf{gen}}$ and accept the samples with probability proportional to $\frac{p_{\mathsf{data}}}{p_{\mathsf{gen}}}$, which is equivalent to sampling from $p_{\mathsf{data}}$. We estimate the density ratio using an optimal discriminator $D^* = \frac{p_{\mathsf{data}}}{p_{\mathsf{data}} + p_{\mathsf{gen}}}$ for a given generator $G$ following the discriminator rejection sampling Azadi et al. (2018). In practice, we use the discriminator $D$ of cGAN as a proxy for $D^*$.

## 4.3 Supervised: Learning the fair distribution

In the supervised scenario, we intervene in the generator to upsample the minority samples described by a fair reference set (weakly-supervised) or labels of sensitive attributes (semi-supervised). We illustrate how to train the cGAN given each form of supervision.

### 4.3.1 Weakly-supervised

**Fairness intervention.** Assuming that we have a reference set $\mathcal{S}_{\mathsf{ref}}$ representing the desired fair distribution, where $p_{\mathsf{fair}}(x|y)$ is the empirical distribution of $\mathcal{S}_{\mathsf{ref}}$, we intervene in the generator to follow the reference set using the density ratio between the reference and data distributions. To achieve this, we train a binary classifier $f_{\mathsf{ref}} : x \mapsto r \in (0,1)$ to distinguish whether a sample belongs to the reference ($r = 1$) or data ($r = 0$) distributions, similar to the density ratio trick used in GANs. The (unnormalized) resampling probability is given by:

$$w(x) = \frac{p(f_{\mathsf{ref}}(x) = 1)}{Mp(f_{\mathsf{ref}}(x) = 0)} \tag{2}$$

where $M > 0$ is some large constant to ensure $w(x) \leq 1$. During training, we draw samples from $p'_{\mathsf{data}}(x) := \frac{w(x)}{W} p_{\mathsf{data}}(x)$, where $W$ is a normalizing factor that makes $p'_{\mathsf{data}}(x)$ a probability density function.

**Corrective sampling.** We train an additional discriminator $D'$ which is responsible for distinguishing between the reference and generator distributions. Once trained, we use rejection sampling (similar to the unsupervised scenario) to correct the samples to align with the desired fair distribution $p_{\mathsf{fair}}(x|y)$.

### 4.3.2 Semi-supervised

**Fairness intervention.** Assuming we have a set of samples with sensitive attribute labels $\mathcal{L} := \{(x, y, a)\}$ and a set of unlabeled samples $\mathcal{U} := \{(x, y)\}$, we first train an attribute classifier $f_a : x \mapsto a$ using the labeled data $\mathcal{L}$, which can also leverage the unlabeled data $\mathcal{U}$ using semi-supervised learning techniques. Using this classifier, we estimate the population of each group $g = (y, a)$ over all samples, constructing the pseudo-labeled dataset $\mathcal{D} := \mathcal{L} \cup \hat{\mathcal{U}}$, where $\hat{\mathcal{U}} := (x, y, \hat{a})$ for $(x, y) \in \mathcal{U}$ and $\hat{a} := f_a(x)$. To balance the occurrence of each group, we intervene with the generator by resampling the training data with a probability inversely proportional to the group population. Specifically, the (unnormalized) resampling probability is

given by:

$$w(x) = \frac{1}{|\{(x', y', a') \in \mathcal{D} : (y', a') = (y, f_a(x))\}|}. \tag{3}$$

As in the weakly-supervised scenario, we draw samples from $p'_{\mathsf{data}}(x) = \frac{w(x)}{W} p_{\mathsf{data}}(x)$ during training.

**Corrective sampling.** Similar to the unsupervised case, we correct the generator distribution to match the desired fair distribution $p_{\mathsf{fair}}(a|y)$ via rejection sampling. However, we cannot simply reuse the discriminator $D$ from cGAN since it is trained with a biased data distribution that is resampled by Eq. (3). Thus, we train an additional discriminator $D'$ to estimate the density ratio between the unbiased data and generator distributions, using $D'$ to recover the original data distribution $p_{\mathsf{data}}(x|y)$. After recovering $p_{\mathsf{data}}(x|y)$, we apply another rejection sampling stage to control the desired group ratio $p_{\mathsf{fair}}(a|y)$ using the predicted attribute $f_a(x)$. This two-stage rejection sampling approach corrects the samples to follow the desired fair distribution.

## 5 Experiments

### 5.1 Experimental setup

**Datasets and models.** We evaluate our framework on the CelebA (Liu et al., 2015) dataset and the Waterbirds (Sagawa et al., 2020) dataset. The CelebA dataset consists of 162,770 face images of celebrities and has 40 attribute annotations. We select two attributes that are spuriously correlated, one as an input condition and the other to evaluate the occurrence ratio of the minority attribute. The Colored MNIST dataset (Arjovsky et al., 2019) consists of 40,000 digit images with binary labels (0 for digits 0-4, 1 for digits 5-9). Each digit is colored either red or green to impose a spurious correlation between binary labels and colors. We do not flip the final label, unlike the original construction. The Waterbirds dataset consists of 4,795 bird images with two types of birds (waterbirds and landbirds) on two types of backgrounds (water and land); bird type is used as an input condition, and the background is used to evaluate the occurrence ratio of the minority attribute. In all our experiments, we use BigGAN (Brock et al., 2019) for the conditional generative models and ResNet-50 (He et al., 2016) for the classifiers. For Colored MNIST experiments, we use ResNetGAN (Gulrajani et al., 2017) for the conditional generative models and a 4-layer convolutional network for the classifiers.

**Evaluation metrics.** We evaluate the conditional generative models in two aspects: (a) sample quality, i.e., $p(x|y)$, and (b) attribute balancing, i.e., $p(a|y)$. For (a) in unsupervised scenarios, we use the intra-class Fr'echet Inception distance (Intra-FID) (Heusel et al., 2017), which measures both sample quality and diversity of the conditional distribution. Specifically, Intra-FID measures the distance between the feature distributions of the data and generated samples. For supervised scenarios, we use a fair version of Intra-FID (Fair Intra-FID), which measures the distance between the fair and generated samples. For (b), we report the occurrence ratio of the minority (or sensitive) attribute, which should follow the ratio of the data and fair distributions for unsupervised and supervised scenarios, respectively. We also report the estimated occurrence ratio for the (training/reference) dataset of the classifier, as the prediction can be under- (or over-) estimated.

**Baselines.** We extend the prior (unconditional) fair generation methods as the baselines for the fair conditional generation, i.e., modify their formula from $p(x)$ to $p(x, y)$. We consider three representative methods: for unsupervised (Lee et al., 2021), weakly-supervised (Choi et al., 2020) scenarios, and supervised (Tan et al., 2020), respectively. For the unsupervised method, we consider Dia-GAN (Lee et al., 2021), which resamples training data based on an estimated log-density ratio to emphasize minority samples. For the weakly supervised method, we consider Choi et al. (2020), giving different weights for each sample based on the density ratio classifier trained to distinguish the training dataset and the reference dataset. For the supervised method, we consider FairGen (Tan et al., 2020), a latent variable modification method based on the attribute classifier trained on the latent variable space. See Appendix B for a detailed explanation.

**Computation time.** Training of the original BigGAN on CelebA for 200k iteration takes 30 hours on a single TITAN Xp GPU and 40 CPU cores (Intel Xeon CPU E5-2630 v4). FICS takes a longer training time due to an additional classifier; takes $\times 1.6$ times for our training setup.

Table 1: **Unsupervised results (CelebA).** Comparison of unsupervised fair conditional generation methods under CelebA, using (a) "blondness" or (b) "wearing lipstick" attributes as a class condition. We report the Intra-FID (↓) and minority occurrence ratio (%) of generated samples. We report the minority occurrence ratio of the training dataset and the estimated values by attribute classifier. Our method achieves the best of both worlds: almost matching the minority occurrence ratio of the original dataset (estimated by the classifier) while producing high-quality samples. The lowest intra-FID and the closest minority occurrence ratio to the estimated are marked in bold.

(a) Blondness

|  | Intra-FID (↓) | | Minor in Blond (%) | | |
|---|---|---|---|---|---|
|  | Blond | Non-blond | Male | Eyeglasses | No lipstick |
| Dataset |  |  | 5.72 | 1.66 | 18.78 |
| Estimated |  |  | 5.85 | 1.85 | 17.61 |
| Unsupervised |  |  |  |  |  |
| cGAN (Mirza & Osindero, 2014) | 1.86 | 1.64 | 3.97 | 0.88 | 11.04 |
| Dia-GAN (Lee et al., 2021) | 1.88 | 1.84 | 5.06 | 1.49 | 14.10 |
| FICS (ours) | **1.78** | **1.38** | **6.13** | **1.90** | **16.34** |

(b) Wearing lipstick

|  | Intra-FID (↓) | | Minor in Lipstick (%) | | |
|---|---|---|---|---|---|
|  | Lipstick | No lipstick | Male | Eyeglasses | Hat |
| Dataset |  |  | 0.58 | 1.05 | 1.25 |
| Estimated |  |  | 0.80 | 1.21 | 1.87 |
| Unsupervised |  |  |  |  |  |
| cGAN (Mirza & Osindero, 2014) | 1.36 | 1.67 | 0.28 | 0.76 | 1.04 |
| Dia-GAN (Lee et al., 2021) | 1.36 | 1.66 | 0.30 | 0.86 | 1.09 |
| FICS (ours) | **1.24** | **1.41** | **0.33** | **1.02** | **1.53** |

Table 2: **Unsupervised results (Colored MNIST, Waterbirds).** Comparison of unsupervised fair conditional generation methods under (a) Colored MNIST using "digit" as a class condition and (b) Waterbirds using "bird type" as a class condition. We report the average value over classes for Colored MNIST and the values of each class for Waterbirds. The lowest intra-FID and the closest minority occurrence ratio to the estimated are marked in bold.

|  | Colored MNIST | | Waterbirds | | Landbirds | |
|---|---|---|---|---|---|---|
|  | Intra-FID (↓) | Minor (%) | Intra-FID (↓) | Minor (%) | Intra-FID (↓) | Minor (%) |
| Dataset |  | 20.00 |  | 5.03 |  | 5.00 |
| Estimated |  | 20.00 |  | 8.72 |  | 8.96 |
| Unsupervised |  |  |  |  |  |  |
| cGAN (Mirza & Osindero, 2014) | 10.31 | 0.16 | 44.68 | 5.52 | 24.73 | 4.69 |
| Dia-GAN (Lee et al., 2021) | 10.20 | 0.05 | 44.32 | 5.54 | 24.80 | 4.66 |
| FICS (ours) | **2.78** | **23.50** | **41.80** | **6.75** | **23.46** | **4.89** |

## 5.2 Unsupervised experiments

We first demonstrate the results in an unsupervised scenario, where the goal is to reduce the discrepancy between the generator and data distributions. For CelebA, we use "blondness" and "wearing lipstick" as input conditions. For each condition, we choose {male, eyeglasses, no lipstick} and {male, eyeglasses, hat} as corresponding minority attributes. We report Intra-FID and minority occurrence ratio (compared to the data distribution) of the baselines and FICS in Table 1 and Table 2.

Table 3: **Supervised results.** Comparison of supervised fair conditional generation methods where "blondness" attribute is used as a class condition. We report the Fair Intra-FID (↓) and minority occurrence ratio (%) of generated samples. The lowest fair intra-FID and the closest minority occurrence ratio to the estimated are marked in bold.

| | Target Male ratio: 30% | | Target Male ratio: 50% | |
|---|---|---|---|---|
| | Fair Intra-FID (↓) | Occurrence of | Fair Intra-FID (↓) | Occurrence of |
| | Blond | Male in Blond (%) | Blond | Male in Blond (%) |
| Reference | | 30.00 | | 50.00 |
| Estimated | | 28.60 | | 47.15 |
| Weakly-supervised | | | | |
| Choi et al. (2020) | 10.71 | 7.60 | 10.70 | 18.34 |
| FICS (ours) | **10.36** | **15.65** | **10.24** | **27.04** |
| Semi-supervised | | | | |
| FairGen (Tan et al., 2020) | 12.71 | **29.54** | 11.98 | 49.76 |
| FICS (ours) | **10.56** | 25.72 | **11.68** | **47.11** |

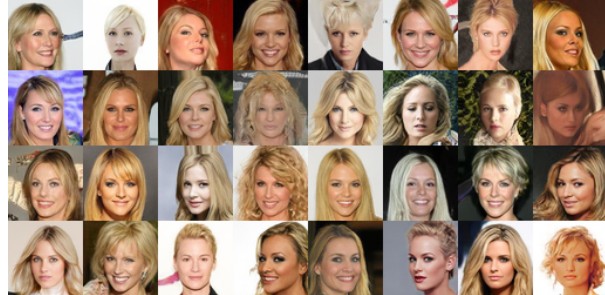 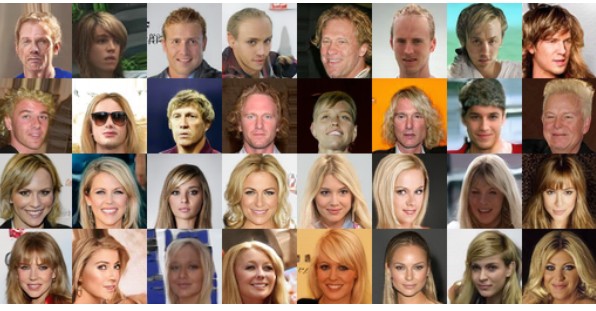

(a) Vanilla BigGAN (Brock et al., 2019)  (b) FICS (ours)

Figure 5: **Visualization.** Blond-conditionally generated samples from the original BigGAN and our method, under the semi-supervised scenario with a target male ratio of 50%. Our method creates diverse blond male images, unlike the original BigGAN is biased toward female images.

In the CelebA dataset, the vanilla cGAN suffers from the spurious causality issue, significantly amplifying the imbalance of minority attributes, e.g., only creates 3.97% of blond males while the original dataset contains 5.85%. Dia-GAN relieves this amplification and increases the minority occurrence; however, it often decreases the Intra-FID, i.e., losing the diversity as a cost of the boosted minority. In contrast, our proposed method, FICS, achieves the best of both worlds: it improves the Intra-FID of all considered cases and recovers the minority occurrence of the data distribution, e.g., producing 6.13% of blond males, almost matching the 5.85% of the training dataset.

We observe a similar tendency in the Colored MNIST and Waterbirds datasets. In the Colored MNIST dataset, both vanilla cGAN and Dia-GAN fails to generate sufficient number of minority group samples. In contrast, FICS achieves a minority occurrence ratio close to the training dataset. In the Waterbirds dataset, the vanilla cGAN suffers from the spurious causality issue, providing only 5.52% and 4.69% of minority samples of waterbirds and landbirds, respectively. Dia-GAN shows similar results to the vanilla cGAN, resulting from the almost identical sample weights. FICS shows improved Intra-FID and the closest minority occurrence to that of the training set for both classes.

### 5.3 Supervised experiments

We demonstrate the results for the supervised scenario, where the goal is to follow fair distribution. We conduct experiments on the CelebA dataset, using "blondness" as an input condition and "gender" as a

Table 4: **Ablation studies.** Comparison of (a) unsupervised and (b) semi-supervised fair conditional generation methods with or without each component of FICS. For (b), we use "blondness" attribute is used as a class condition and 50% male in blond as the desired distribution. The lowest fair intra-FID and the closest minority occurrence ratio to the estimated are marked in bold. Both fairness intervention (FI) and corrective sampling (CS) benefits.

| | (a) Unsupervised | | | (b) Semi-supervised | |
| | Intra-FID ($\downarrow$) | Male in Blond (%) | | Fair Intra-FID ($\downarrow$) | Male in Blond (%) |
|---|---|---|---|---|---|
| cGAN | 1.86 | 3.97 | | 13.16 | 3.97 |
| FI only | 1.80 | 6.02 | | 12.30 | 18.23 |
| CS only | 1.85 | 5.02 | | 12.91 | 38.49 |
| FICS (ours) | **1.78** | **6.13** | | **11.68** | **47.11** |

sensitive attribute. Recall that the original data distribution has only a few blond males; we aim to learn a fair distribution that produces {30%, 50%} of male images conditioned on blond. We report the Fair Intra-FID and minority occurrence ratio (should be similar to the desired fair distribution) of the baselines and our proposed framework. Table 3 shows the results on weakly- and semi-supervised scenarios: we will discuss the details and observations in the remaining section.

**Weakly-supervised.** We use the same 4,000 samples (but without sensitive attribute annotations) as the reference set for the weakly-supervised setup. We compare our method with Choi et al. (2020), and use the same binary classifier that distinguishes the reference and data distributions for both methods; recall that we resample training data with the classifier while Choi et al. reweights the loss. The results show that our method outperforms Choi et al. for all considered cases, both Fair Intra-FID (i.e., generation quality) and minority occurrence ratio (i.e., attribute balancing). Here, both methods still have a gap with the semi-supervised methods since they rely on the weaker form of supervision. Still, we remark that our method significantly reduces the gap between semi-supervised and weakly-supervised scenarios, particularly in terms of the minority occurrence ratio.

**Semi-supervised.** We use 4,000 labeled (sensitive attribute) samples with the same attribute occurrence ratio of the desired fair distribution for the semi-supervised setup. We compare our method with FairGen, and use the same attribute classifier for both methods, which is trained by a semi-supervised learning technique called FixMatch (Sohn et al., 2020). The results show that our method shows better Fair Intra-FID than FairGen, while both methods show comparable minority occurrence ratio. Recall that FairGen modifies the prior distribution to generate minority samples; it distorts the prior to the low-density area of the original Gaussian distribution. Thus, FairGen often generates low-fidelity samples (though satisfying the desired attribute), leading to the lower FID.

**Visualization.** Figure 5 visualizes the generated samples conditioned on blond from vanilla BigGAN and debaised one by FICS. We use the semi-supervised version with a target male ratio of 50% for our method. Unlike the original BigGAN mostly creates female images, our method creates diverse blond male images.

## 5.4 Ablation study

We conduct an ablation study to see the contribution of each component in our semi-supervised method. The result is shown in Table 4. With fairness intervention during the training (based on pseudo-labeling and group balancing), we could observe gain in both fair intra-FID and minority occurence ratio, still there exists large gap between the occurrence ratio achieved and the desired occurrence ratio. With corrective sampling after the training, we could observe reasonable occurrence ratio for male, still there is room to be improved. With full components of our semi-supervised method, we can see further improvement in both minority occurrence ratio and fair intra-FID.

# 6 Conclusion

We highlight the issue of spurious causality of conditional generation, an important yet underexplored fairness problem in conditional generation. To address this issue, we propose the FICS framework, which leverages intervened training and corrected sampling. We hope that our work inspires further research in spurious correlation and generative models, particularly the spurious causality in text-to-image generative models.

**Broader Impact Statement**

Although we alleviate some spurious causality of conditional generative models, the model can still retain unrecognized biases. Since the algorithmic debiasing cannot remove all possible biases from the model, the users should carefully check the model on their usage.

**Acknowledgments**

This work was partly supported by Institute of Information & communications Technology Planning & Evaluation (IITP) grant funded by the Korea government (MSIT) (No.2019-0-00075, Artificial Intelligence Graduate School Program (KAIST); No. 2022-0-00184, Development and Study of AI Technologies to Inexpensively Conform to Evolving Policy on Ethics), Korea Evaluation Institute of Industrial Technology (KEIT) grant funded by the Korea government (MOTIE), and the National Research Foundation of Korea (NRF) grant funded by the Korea government (MSIT) (2022R1F1A1075067).

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

# A Implementation details

## A.1 Classifiers

**Network architectures.** For CelebA and Waterbirds, we use the `torchvision` implementation of ResNet-50 starting from ImageNet-pretrained weights. For Colored MNIST, we use 4-layer convolutional network with channel sizes of 32, 32, 64, 64, kernel size of 5, 3, 3, 3, and stride of 2, 1, 2, 1.

**Biased classifiers for FICS.** It is well-known that ERM trained classifier trained with a biased dataset inherits spurious correlations existing in the dataset (Sagawa et al., 2020; Liu et al., 2021). Thus, we did not perform any additional technique to amplify the spurious correlation. To handle a class imbalance in the dataset, we use group DRO (Sagawa et al., 2020) to ensure low worst-class error. We use the SGD optimizer with a learning rate of 0.001, momentum 0.9 for 100 epochs with batch size 64. In addition, we use $\ell_2$ regularization of 0.0005 and early stopping for the best validation accuracy.

**Attribute classifiers for evaluation.** For the attribute classifiers for evaluation, we train the models with group DRO to ensure low worst-group error. We use the SGD optimizer with a learning rate of 0.0001, momentum 0.9 for 15 epochs with batch size 64. In addition, we use $\ell_2$ regularization of 0.01 and early stopping for the best validation accuracy. Table 5, 6 reports the accuracy of attribute classifiers on the CelebA and Waterbirds dataset, respectively. The attribute classifier for classifying color on the Colored MNIST dataset showed 100% accuracy for all groups.

Table 5: Accuracy of the attribute classifiers used for evaluation on the CelebA dataset.

| Attribute of interest | Blond | | Non-blond | |
|---|---|---|---|---|
| | Present | Absent | Present | Absent |
| Male | 86.11 | 97.18 | 97.20 | 96.08 |
| Eyeglasses | 96.77 | 99.30 | 92.91 | 99.56 |
| Lipstick | 82.47 | 83.79 | 86.83 | 92.10 |

Table 6: Accuracy of the attribute classifiers used for evaluation on the Waterbirds dataset.

| Attribute of interest | Waterbirds | | Landbirds | |
|---|---|---|---|---|
| | Water | Land | Water | Land |
| Background | 94.74 | 90.98 | 89.70 | 95.93 |

**Reference set classifiers for weak-supervision.** For the reference set classifiers, we follow the details from the original paper (Choi et al., 2020). We use the Adam optimizer with a learning rate of 0.0001 without any $\ell_2$ regularization for 15 epochs with batch size 64. We use the same reference set classifiers for the weakly-supervised baseline Choi et al. (2020) proposed and our weakly-supervised version. We use $\alpha = 0.3$ for target male ratio 30% and $\alpha = 0.5$ for target male ratio 50% for the weakly-supervised baseline.

**Attribute classifiers for semi-supervision.** We use 4,000 samples with the sensitive attribute annotation and the rest of the samples in the training set without the sensitive attribute annotation to train the sensitive attribute classifier. We use FixMatch (Sohn et al., 2020), a recent SOTA semi-supervised learning technique, to train the sensitive attribute classifier. We only use random horizontal flip for data augmentation and use a fixed threshold of 0.95 for $\tau$. We use the SGD optimizer with a learning rate of 0.001, momentum 0.9 for 100 epochs with batch size 64. In addition, we use $\ell_2$ regularization of 0.005 and early stopping for the best validation accuracy.

## A.2 Generative models

**Network architectures.** For generative model, we implement our code based on `PyTorch-StudioGAN` repository[2]. We use BigGAN for the CelebA and Waterbirds experiments, and ResNetGAN with spectral normalization (Miyato et al., 2018) for the Colored MNIST experiments. For CelebA and Colored MNIST, we train our generative model from scratch. For Waterbirds, we train our model starting from ImageNet-pretrained weights provided by `PyTorch-StudioGAN` repository.

**Training details.** For all the baselines and our method on the CelebA dataset, we train BigGAN with the same configurations. We train the models using the Adam optimizer with learning rate of 0.0002, $\beta_1 = 0$, $\beta_2 = 0.999$, for total 200k iterations with batch size 32. We update the discriminator 4 times per one generator step. For our Waterbirds experiments, we train models using the Adam optimizer with learning rate of 0.0001 for the generator and 0.0002 for the discriminator, $\beta_1 = 0$, $\beta_2 = 0.999$, for total 20k iteration with batch size 128. We update the discriminator 2 times per one generator step. We use DiffAug (Zhao et al., 2020) to supplement the limited number of training data. For the Colored MNIST experiments, we train the models using the Adam optimizer with learning rate of 0.0002, $\beta_1 = 0.5$, $\beta_2 = 0.999$, for total 20k iterations with batch size 64. We update the discriminator 5 times per one generator step.

**Dia-GAN.** For the CelebA dataset, we train 150k iterations for phase 1 and use log density ratios from 140k to 150k iterations to reweight samples for phase 2. For the Waterbirds dataset, we train 5k iterations for phase 1 and use log density ratios from 500 to 5 iterations to reweight samples for phase 2. For the Colored MNIST dataset, we train 4k iteration for phase 1 and use log density ratios from 100 to 4k iterations to reweight samples for phase 2. Following the original paper, we clip the sample weights from 0.01 to 0.5.

**FICS.** For the CelebA dataset, we first pretrain 100k iterations with the vanilla configurations and then finetune for the next 100k iterations. For finetuning, we freeze the discriminator until 4 layers using FreezeD (Mo et al., 2020) and add the proposed regularizer. We also start to resample after 100k iterations when supervision is provided. For the Waterbirds and Colored MNIST dataset, we did not perform any additional finetuning before FICS training. For the $\lambda$ of FICS regularizer, we tuned over $\{0.002, 0.005, 0.01, 0.02\}$ and use the values with the best FID.

# B Baselines

We extend the prior (unconditional) fair generation methods as the baselines for the fair conditional generation, i.e., modify their formula from $p(x)$ to $p(x, y)$. We consider three representative methods for unsupervised (Lee et al., 2021), supervised (Tan et al., 2020), and weakly-supervised (Choi et al., 2020) scenarios.

**Unsupervised.** Unsupervised fair generation aims to promote underrepresented samples and reduce the gap between data and generator distributions. For instance, Dia-GAN (Lee et al., 2021) resamples the training data based on the log-density ratio $\log p_{\mathsf{data}}(x)/p_{\mathsf{gen}}(x)$ estimated by the discriminator.

**Supervised.** Given annotations of sensitive attribute $a \in \mathcal{A}$, a straightforward solution for a fair conditional generation is conditioning both $y$ and $a$, as one can explicitly control the generated attributes using $p(x|y, a)$. However, this joint-condition approach is hard to train when the data is imbalanced; too scarce for the minority group $(y, a)$. As an alternative, FairGen (Tan et al., 2020) propose a latent modification approach that modifies an unconditional generative model to a conditional model by learning a prior distribution $p_a(z)$ that restricts the samples under $p(x|a)$, unlike the original prior $p(z)$ sampling $p(x)$. Note that the supervised method performs well if the target attribute $a$ is known; however, it requires heavy annotation costs and cannot prevent the unknown biases.

**Weakly-supervised.** Choi et al. (2020) consider a weaker form of the fairness supervision, assuming a reference set $\mathcal{S}_{\mathsf{ref}}$ with empirical distribution $p_{\mathsf{ref}}(x) \approx p_{\mathsf{fair}}(x)$. Specifically, they reweighted the training data with density ratio $p_{\mathsf{ref}}(x)/p_{\mathsf{data}}(x)$ estimated by a classifier distinguishing the reference and data distributions. Note that the weakly-supervised method heavily depends on the collection of the reference set; it is often unrealistic to collect the oracle reference set in practice.

---

[2] https://github.com/POSTECH-CVLab/PyTorch-StudioGAN

## C    Additional experiments

### C.1    The relationship between the biased classifier and the generator

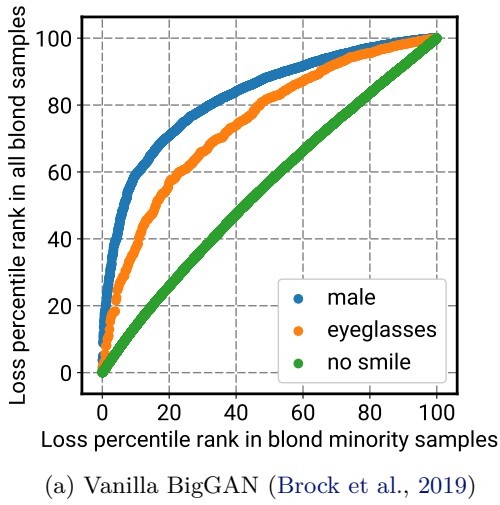

(a) Vanilla BigGAN (Brock et al., 2019)

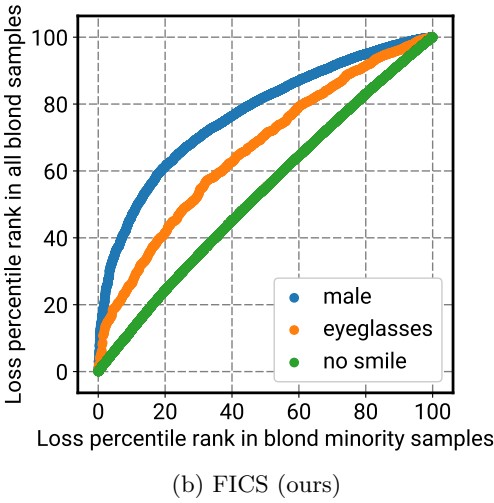

(b) FICS (ours)

Figure 6: The relationship between the biased classifier and the generator represented through classification loss percentile ranks of generated samples. Given a point, its x, y coordinate represents the percentile rank of its classification loss among blond minority samples and all blond samples respectively. The correlation between the classification loss and the chance to be minority still remains after the FICS training.

To validate the effectiveness of our proposed regularization approach, we conducted an investigation into the correlation between the classification loss and the probability of a sample being a minority at the end of the FICS training. We present the results in Figure 5, where we plot the classification loss percentile ranks as shown in Figure 3. As the number of minority samples generated by the FICS training increased, we observed that a sample with a loss ranking in the 20th percentile among blond male samples had a relatively lower rank in the FICS-trained cGAN compared to the vanilla cGAN. However, we found that the correlation between the classification loss and the likelihood of a sample being a minority persisted even after the FICS training. Based on these findings, we can conclude that our regularizer, which is based on the observed correlation, consistently encouraged the cGAN to generate minority samples throughout the entire training process.

