# OpenReview forum: "Breaking the Spurious Causality of Conditional Generation via Fairness Intervention with Corrective Sampling"
_TMLR — Accepted by TMLR_

### Review · Reviewer_HYiU · 2023-04-09

**Summary Of Contributions:**

This paper studies the mitigation of conditional generative models in an unsupervised setting where the sensitive attribute is unknown. The idea is to build a classifier separately on the training data and use the classification loss to indicate underrepresented samples. Then, a regularizer is added to the loss of the generative model to promote the generation of samples with large classification losses. Finally, rejection sampling is used to recover the original distribution.

**Audience:**

Yes

**Broader Impact Concerns:**

None.

**Claims And Evidence:**

No

**Requested Changes:**

It would be helpful for comprehension if the authors can add causal graphs in Section 3 to explain the difference between spurious causality and spurious correlation.

**Strengths And Weaknesses:**

Strength

The paper studies the unsupervised setting which is different from prior work on fair generation where the sensitive attribute is given. The idea of using the classification loss to infer underrepresented samples is reasonable.

Weakness

To explain the motivation of the paper, the authors introduce the concept of spurious causality, which is not quite convincing to me. To create a causal path from $Y$ (the act of conditioning on the label) to $A$ (the sensitive attribute), the authors assume that $A$ is a function of $X$. However, in reality we often assume that $X$ is a function of $A$ since sensitive attributes like gender and race represent the inherent natures of individuals. In addition, the paper does not explain why spurious causality is more problematic to downstream classifiers than spurious correlations.

The proposed method FICS is actually decoupled with the proposed spurious causality concept and is directly based on the observation that “the generated samples that are classified as the minority group are more likely to have higher classification loss”.

---

> ### Author Response · Authors · 2023-04-25
> **Response to Reviewer HYiU**
>
> We sincerely appreciate your efforts and insightful comments to improve the manuscript. We respond to each of your comments one-by-one in what follows. In the revised manuscript, we have marked the revisions with “green.”
>
> -------------------
>
> **The concept of spurious causality**
>
> We note that we do not refer to any causal structure that lies on the data, and our new concept of the spurious causality is only meaningful within the context of learned cGAN models.
> In the context of cGAN models, users apply a conditioning operation on Y to generate X. In this framework, we present A as a function of X, where a(x) captures the perception of individuals recognizing attribute A based on the generated data X. As a result, there is a potential for the interpretation of spurious causality Y -> A due to the perception of A based on generated X, which could lead to a misperception among users.
>
> -------------------
>
> **Why spurious causality is more problematic**
>
> Sorry for making confusion. We revised our explanation to clarify why spurious causality is a more significant issue than spurious correlation from the users' perspective instead of the downstream classifier.
>
> -------------------
>
> **Design of the proposed method**
>
> Our proposed concept of spurious causality pertains to the potential misconceptions that users of cGAN models may have. We note that our proposed method is not based on any assumed causal model or causality method, but rather aims to mitigate the spurious correlation in the distribution of generated results to eliminate any misconceptions that users may have regarding Y -> A. Our proposed FICS includes not only the method based on the aforementioned observation, but also resampling for fairness intervention and corrective sampling processes aimed at ensuring that the generated results follow the reference distribution based on the supervision provided.
>
> -------------------
>
> **Causal graphs explaining spurious causality**
>
> Thank you for your suggestion. We have added causal graphs that illustrate how spurious causality can arise in the presence of spurious correlation in the data in Section 3.

---

### Review · Reviewer_9tsi · 2023-04-11

**Summary Of Contributions:**

This work aims to solve the unfairness/spurious correlations caused by data imbalance in conditional generative models. For example, BigGAN conditioning on the attribute "blond" will generated female samples more frequently. It formulates the problem as approximate p(a|y) from the original dataset. It proposes different corrective sampling methods for different settings. Experiments show the proposed method can achieve high quality as well as mitigate the imbalance issue in conditional generation.


**Audience:**

Yes

**Claims And Evidence:**

Yes

**Requested Changes:**

- In the unsupervised setting, the authors use a classifier to classify minority and majority groups, how did they train this classifier and how this is unsupervised if a classifier is trained in a supervised manner?

- Can the authors explain, in the rejection sampling of unsupervised setting, why the samples have to be accepted with the probability proportional to p_{data}/p_{gen}?

- Is eq.(2) correct? It looks to me the RHS is not a probability density function.

- Does FICS always take the same training time under different settings (unsupervised, weakly supervised and semi supervised).

- The authors did not specify how w(x) will be used in resampling.

**Strengths And Weaknesses:**

S
- This work investigates an important fairness problem -- the amplified bias (imbalance) in conditional generative models.
- This work covers the setting of unsupervised, weakly supervised and semi supervised learning.

W
- It is not clear, from eq.(2) and the paragraphs following it, why do conditional generative models always amplify the imbalance conditioned on certain attribute in the dataset.

- Although empirically, with a cGAN trained with the original loss, authors find the imbalance in sampling is correlated with the classification loss. Note that this correlation is a property of the cGAN model trained with original loss. But it is not clear why training the cGAN model with eq.(3) can guarantee that more balanced samples would be generated.

- The presentation can be improved.

---

> ### Author Response · Authors · 2023-04-25
> **Response to Reviewer 9tsi**
>
> We sincerely appreciate your efforts and insightful comments to improve the manuscript. We respond to each of your comments one-by-one in what follows. In the revised manuscript, we have marked the revisions with “green.”
>
> -------------------
>
> **Amplification of imbalances**
>
> We clarify that the “amplification of imbalance” is an empirically observed phenomenon; we do not intend to deduce the phenomenon in any sense through eq.(2) and the subsequent paragraphs. We note that this phenomenon has been also previously observed empirically in classification literature.
>
> -------------------
>
> **Motivation of eq.(3)**
>
> We have designed our method to address the imbalance in sampling by promoting cGANs to generate samples with high classification loss. This is due to our observation of the correlation between chance to be minority and classification loss, which suggests that samples with high classification loss are likely to be minority samples. This correlation consistently holds whenever the cGAN model is trained with original loss or the objective function in eq.(3). We add the relationship between the biased classifier and the generator trained with eq.(3).  in Appendix C1.
>
> -------------------
>
> **Classifier for majority/minority groups**
>
> In our unsupervised setting, we lack supervision with regards to the sensitive attribute label. However, we assume that we have access to the class-condition for every data point we have. The classifier uses the class-condition as a label and is trained to classify it.
>
> -------------------
>
> **Acceptance probability of rejection sampling**
>
> Thank you for your suggestion, we have added a more detailed explanation on the rejection sampling of unsupervised setup in the corresponding section.
>
> -------------------
>
> **Correctness of eq.(2)**
>
> Since A is a discrete random variable, we use 1[a = a(x)] as a probability mass function of A conditioned on X, Y. Therefore, RHS of eq.(2) is \int_X p(a|x,y)p(x|y) dx = \int_X (a,x|y) dx = p(a|y).
>
> -------------------
>
> **Training time**
>
> For all settings (unsupervised, weakly supervised, semi-supervised),  FICS takes the same amount of time per iteration.
>
> -------------------
>
> **Use of w(x)**
>
> Sorry for the misspecification. We clarified how w(x) is used for resampling in the training phase.

---

### Review · Reviewer_4jmF · 2023-04-11

**Summary Of Contributions:**

The authors introduce FICS, a generative model that learns a fair distribution (wrt a protected feature) of the data. In particular, fairness is defined as being robust against "spurious causality" whereby a model may wrongly learn relationships such as hair colour -> gender (the example being blonde -> female).

FICS learns a fair distribution through an adversarial term in their loss function where the distribution is corrected by "worsening" the ability to classify a protected attribute from the representation.

**Audience:**

Yes

**Broader Impact Concerns:**

I don't think there are any BI concerns.

**Claims And Evidence:**

Yes

**Requested Changes:**

Please consider the weaknesses I listed above and provide a new insight/answer to rebut those.

Some additional questions:

- How would this approach work on tabular data? Causality is definitely more evolved in a tabular domain.

- Couldn't you just use _any_ generative model in combination with subsampling?

**Strengths And Weaknesses:**

STR.

- the authors acknowledge the importance of causality in a fairness definition and use it to their benefit. I like this perspective and would advocate more research on this topic.

- It seems the proposed method can be used post-hoc after training other generative models (though this needs confirming -> I asked a question in this vain below)


WKN.

- The proposed idea seems quite related to counterfactual fairness which should probably be discussed and compared against since it is quite a popular paper (https://proceedings.neurips.cc/paper/2017/hash/a486cd07e4ac3d270571622f4f316ec5-Abstract.html)

- following the above, there are quite some relevant papers missing from the authors' comparison (some of them also casual GANs which are trained to promote fair sampling). I listed them below.

- on p4 the authors introduce the notion of "spurious causality", whereby a generative model introduces a causal link based on supriously correlated variables. I don't understand why we need such new terminology, what the authors describe as spurious causality is still spurious correlation? why would this be different? Nothing in a GAN makes a correlation causal, just as there is nothing in a GAN that makes a spurious correlation spuriously causal.

- Fig 2 shows that we sample from the model by first performing a do-operation. Later in sec. 4.3.2 we see that the authors translate this do operation by simply weighting the likelihood of each sample, such that the complete sample matches a fair distribution. How is this causal? Especially when p^{do} is given (in the form of a reference distribution), I'm completely at a loss why the proposed model is in any way causally more aware than subsampling from another model.

- There are more fairness notions than simple class imbalance (see the refs I listed below), how would this model translate those into a new ref. distribution? Furthermore, am I correctly understanding that the model requires retraining whenever we want to update our fairness definition?


### Refs.

van Breugel, B., Kyono, T., Berrevoets, J., & van der Schaar, M. (2021). Decaf: Generating fair synthetic data using causally-aware generative networks. Advances in Neural Information Processing Systems, 34, 22221-22233.

Lu Zhang, Yongkai Wu, and Xintao Wu. A causal framework for discovering and removing direct and indirect discrimination. In Proceedings of the Twenty-Sixth International Joint Conference on Artificial Intelligence, IJCAI-17, pages 3929–3935, 2017. doi: 10.24963/ijcai. 2017/549. URL https://doi.org/10.24963/ijcai.2017/549.

Flavio P. Calmon, Dennis Wei, Karthikeyan Natesan Ramamurthy, and Kush R. Varshney. Optimized data pre-processing for discrimination prevention, 2017

Michael Feldman, Sorelle A Friedler, John Moeller, Carlos Scheidegger, and Suresh Venkata subramanian. Certifying and removing disparate impact. In proceedings of the 21th ACM SIGKDD international conference on knowledge discovery and data mining, pages 259–268, 2015.

Rich Zemel, Yu Wu, Kevin Swersky, Toni Pitassi, and Cynthia Dwork. Learning fair representations. In International conference on machine learning, pages 325–333. PMLR, 2013.

Depeng Xu, Yongkai Wu, Shuhan Yuan, Lu Zhang, and Xintao Wu. Achieving causal fairness through generative adversarial networks. In Proceedings of the Twenty-Eighth International Joint Conference on Artificial Intelligence, 2019.

---

> ### Author Response · Authors · 2023-04-25
> **Response to Reviewer 4jmF**
>
> We sincerely appreciate your efforts and insightful comments to improve the manuscript. We respond to each of your comments one-by-one in what follows. In the revised manuscript, we have marked the revisions with “green.”
>
> -------------------
>
> **Comparison to counterfactual fairness**
>
> Our framework shares some similarities with counterfactual fairness in terms of its capability to conduct counterfactual analysis on specific attributes. However, our framework performs do-operation on the class attribute and focuses on the distribution of the sensitive attribute in the generated images. On the other hand, counterfactual fairness aims to achieve fair prediction by performing do-operation on the sensitive attribute. As such, the methods used to achieve counterfactual fairness are not directly applicable to resolving the problem of interest in our framework. We included the above discussion in our related work section.
>
> -------------------
>
> **Comparison to causal GANs**
>
> We would like to clarify that our paper focuses on the fairness issue in conditional GAN models specifically in the vision domain, thus we have limited the scope of our comparison accordingly. The suggested reference causal GAN papers mainly focus on the tabular domain, where generating tabular data following a causal graph is the primary objective, assuming prior knowledge of the causal structure among the attributes. Unlike the tabular domain, in the vision domain, it is not typical for each data point to have all attribute labels, due to the high cost associated with manually annotating each attribute.
>
> In response to the suggestion, we attempted to train a GAN with the following a causal structure of Y, A -> X as the proposed approach, but our experimental results were not satisfactory due to insufficient minority samples with the y, a combination in our setup. (Specifically, we observed mode collapse issues with the blond, male class.)
>
> We included the above discussion in our related work section.
>
> -------------------
>
> **Adopting new terminology**
>
> Although there is no causal relationship between the class attribute Y and the sensitive attribute A, conditioning the class to generate data can impact the sensitive attribute of the generated data. Consequently, cGAN users who observe the generated data and its sensitive attribute may falsely perceive that manipulating the class attribute alters the sensitive attribute of the generated data, as long as a spurious correlation exists between the class attribute and the sensitive attribute. We refer to this phenomenon as spurious causality of conditional generation, as there is no causal relationship in the underlying data generating process, but users may perceive a causal relationship exists. We have added an explanation of our new terminology in Figure 2.
>
> -------------------
>
> **About do-operation and causal awareness**
>
> Our method focuses on reducing unfairness related to the sensitive attribute resulting from the “do-operation” in the cGAN model, rather than generated samples following a given causal structure. Here, the "do-operation" refers to the act of conditioning on the class to generate images.
>
> -------------------
>
> **Translation to other fairness notions**
>
> Translation to other fairness notions can be accomplished with or without retraining. We may adjust the fairness objective for both fairness intervention and corrective sampling. While sampling alone may be sufficient to improve fairness performance, retraining can further enhance the performance. For instance, Table 4 shows that applying the corrective sampling alone can improve fairness, but incorporating the fairness intervention, which requires retraining, yields even better results.
>
>
> -------------------
>
> **FICS on tabular data**
>
> Since FICS is designed to address the absence of sensitive attribute annotation in a dataset, it may not be perfectly suited for the tabular domain where sensitive attribute annotation is typically available. However, in situations where the sensitive attribute is not annotated, FICS can serve as a potential remedy.
>
> -------------------
>
> **Extension to other generative models**
>
> While subsampling can be used in combination with any generative model, the regularization using the classifier and rejection sampling in our proposed method are specifically designed to achieve fairness in conditional GAN. Extending our framework to other conditional generative models (such as diffusion models) could be an important direction for future research.

---

### Decision · Action_Editors · 2023-06-05

**Recommendation:** Accept with minor revision

**Comment:**

### Summary

The paper identifies the problem of spurious correlations in conditional generative models (such as cGANs). The authors term this problem as "spurious causality", as users may use such models to make causal deductions. They propose solutions to this problem for both the unsupervised and supervised setting, by minimising the effect of such spurious correlations on the cGAN objective.

### Reviewer Recommendations

Two reviewers found the paper to be interesting, and were overall supportive of acceptance. A third reviewer had concerns on the usage of the term "spurious causality", since this does not refer to any causal structure in the original data, and the techniques proposed are about mitigating spurious correlation (and do not use any techniques from causality). I note that one of the positive reviewers had also raised the question of the difference between spurious correlation and causality.

### AE Recommendation

From my review of the paper, I find the points raised by all reviewers to be valid: the paper tackles what appears to be a novel problem, offers solutions for a few different settings, and shows good empirical results. It is also well-written. At the same time, the notion of "spurious causality" appears to be equivalent to the established notion of "spurious correlation", but specialised for the setting of conditional generative models. While I do understand the authors' motivation for using the former, there are some reasons to argue against it:
- as two reviewers were unclear on this point, it is plausible that other members of the TMLR audience might also be confused about the precise distinction to regular spurious correlations.

- the methods proposed are entirely in line with the spurious correlations literature, and are not apparently generalisable to other causality problems.

- in the precise examples used to evaluate the method, it is not immediately clear that an end-user might make causal inferences based on the suggested interventions. e.g., on Waterbirds, it is not apparent that an end-user might make inferences about the type of bird backgrounds affecting the background. Even on CelebA, it is not obvious that one might infer that "Blondness" _causes_ "Eyeglasses". (I understand that there may be other scenarios where such causal inference could be more plausible.)

On the other hand, if the problem were simply referred to as "spurious causality for generative models", it would not affect any of the technical details of the methods and experiments. While perhaps more verbose, even if one were to retain usage of the term "spurious causality", it would be appropriate to qualify that it is "spurious causality for conditional generation". This is because one could imagine encountering a problem of spurious causality in regular (i.e., non-generative) classification problems, where one attempts to make causal inferences using techniques from the causality literature.

Our recommendation is thus for the paper to be accepted, subject to a softening of the usage of the term "spurious causality". Our suggestion is for the authors to primarily use a term such as "spurious causality for generative models", or a suitable abbreviation. (The discussion on causality could perhaps be contained in one or two focussed sub-sections.) While the term "spurious causality" is used at various points in the paper, the total usage is about 20 times; thus, we believe this terminology change can be accommodated in a minor revision to be reviewed by the AE.


**Audience:**

Reviewers generally agree the audience would be interested in the paper's findings. I agree with this assessment.

**Claims And Evidence:**

Generally good, though the usage of the term "spurious causality" can be confusing. See detailed comments.